# Comparison of Metabolites and Gut Microbes between Patients with Parkinson’s Disease and Healthy Individuals—A Pilot Clinical Observational Study (STROBE Compliant)

**DOI:** 10.3390/healthcare10020302

**Published:** 2022-02-04

**Authors:** Cheol-Hyun Kim, Jeeyoun Jung, Young-ung Lee, Kwang-ho Kim, Sunny Kang, Geon-hui Kang, Hongmin Chu, Se-Young Kim, Sangkwan Lee

**Affiliations:** 1Department of Internal Medicine and Neuroscience, College of Korean Medicine, Wonkwang University, Iksan 54538, Korea; lambroskch@gmail.com (C.-H.K.); www8744@naver.com (Y.-u.L.); bzkimkh@naver.com (K.-h.K.); rkdsunny@gmail.com (S.K.); hongminchu2@gmail.com (H.C.); 2Stroke Korean Medicine Research Center, Wonkwang University, Iksan 54538, Korea; sya0474@naver.com (G.-h.K.); kimse0@hanmail.net (S.-Y.K.); 3KM Science Research Division, Korea Institute of Oriental Medicine, Daejeon 34054, Korea; jjy0918@kiom.re.kr; 4Hanbang Cardio-Renal Syndrome Research Center, College of Oriental Medicine, Wonkwang University, Iksan 54538, Korea

**Keywords:** gut microbes, metabolites, Parkinson’s disease

## Abstract

Introduction: Even if levodopa, dopamine agonists, and others are used for patients with Parkinson’s disease, the effect is not sustained, and side effects such as motor fluctuation and dyskinesia are more likely to appear as the dose increases. Thus, new approaches for managing Parkinson’s disease are needed. This study aimed to compare the metabolites and gut microbes between patients with Parkinson’s disease and healthy individuals. Methods: This was an observational study with a case-control design. Metabolite and gut microbial analyses were performed using blood and stool samples collected from the subjects. Results: Among the metabolites, the acetate, citrate, methionine, and trimethylamine levels were significantly different between the two groups. In the gut microbes, abundance of *Bacteroidetes*, *Prevotella*, *Phascolarctobacterium*, *Pseudoflavonifractor*, *Eisenbergiella*, and *Gemella* were also significantly different between the two groups. Discussion: Metabolites are the products of gut microbes. Therefore, when the gut microbes change, the metabolites change accordingly. Metabolites and gut microbes that were significantly different between the two groups were mostly those involved in lipid and glucose metabolism. Our data may be helpful for the development of new drugs targeting metabolites and gut microbes through large-scale studies in the future.

## 1. Introduction

Parkinson’s disease (PD) is the second most common neurodegenerative disease worldwide [1]. One percent of the population aged >60 years have PD, and its prevalence continues to increase [2]. Although PD has such a high prevalence and gradually worsens the quality of life of PD patients, there is still no clear treatment [3,4]. Even if levodopa, dopamine agonists, and others are used, the effect is not sustained, and side effects, such as motor fluctuation and dyskinesia are more likely to appear as the dose increases [5]. Several studies have reported candidates for neuroprotective agents such as vitamin D, beta-carotene, and bee venom that have the potential to replace conventional treatments, but the evidence is still lacking [6,7]. Therefore, new approaches to PD are needed. 

Gut microbes and metabolites are currently in the spotlight as targets for new therapeutic approaches to PD [8]. There are hundreds of trillions of gut microbes in the human body, and they have a significant impact on normal physiology by producing metabolites [9,10]. Thus, imbalances in gut microbes and metabolites could be associated with neurologic disorders including PD [9]. In support of this hypothesis, Mertsalmi et al. reported that the use of certain antibiotics could increase the risk of PD by altering the gut microbes [11]. Houser and Tansey suggested in their study that the pathogenesis of PD originates from the gut [12]. Initial inflammation in the gut triggers changes in gut microbes, leading to changes in metabolites related to inflammation, which induces neuroinflammation by increasing the blood–brain barrier permeability, ultimately inducing neurodegeneration [12]. In addition, Sun et al. reported that gut microbial dysbiosis is observed in PD patients, and that microbial metabolites and products contribute to the PD pathology [13]. Considering these studies, drugs targeting gut microbes or metabolites could become new therapeutic candidates for PD. 

The purpose of this study was to compare the differences in gut microbes and metabolites between patients with PD and healthy individuals to contribute to the existing research data to select a new therapeutic drug for PD in the future. Although the number of subjects was small, we report our findings because we obtained significant results. 

## 2. Materials and Methods

### Study Design

This research work was an observational study with a case-control design. 

## 3. Subjects

### 3.1. Sample Size Calculation

Our investigation is a pilot study, and we could not find previous data indicating the sample size needed to yield significant results. For the pilot study, a sample size of 20–40 participants was suggested by Kieser and Wassmer [14]. The healthy control (HC) and PD groups were recruited through posters in communities and hospitals from December 2018. The HC group was recruited by matching the age and sex of the PD group. 

### 3.2. Inclusion and Exclusion Criteria for the PD Group

The inclusion criteria were as follows: those diagnosed with PD and taking PD-related drugs who agreed to participate in this study and voluntarily signed the informed consent form and those who consumed traditional Korean dishes such as rice and kimchi. 

The exclusion criteria were as follows: those diagnosed with diseases that may affect the results of this study such as diabetes mellitus (DM) and autoimmune diseases; those taking antibiotics or probiotics; those who regularly drank alcohol and smoked; those unable to collect blood or stool; and those judged to be unsuitable to participate in this study by medical staff. 

### 3.3. Inclusion and Exclusion Criteria for the HC Group

The inclusion criteria were as follows: those who had no underlying disease and were not taking any drugs who agreed to participate in this study and voluntarily signed the informed consent form. Those who were judged to be unsuitable to participate in this study by medical staff were excluded from the study.

## 4. Variables

The variables were metabolites analyzed from the collected blood specimens and the gut microbes analyzed from the collected stool specimens. 

### 4.1. Metabolite Analysis

#### 4.1.1. Blood Collection Method

After collecting 5 mL of blood using the injection needle provided in the blood collection kit, the blood was divided into 3.0 and 2.0 mL samples and packed in serum separate tube and nonautologous-pooled human plasm containers, respectively. Then, the serum and plasma were separated. 

#### 4.1.2. Metabolite Analysis Method

The total of 250 μL of serum was mixed with 500 μL saline solution (10% D_2_O for lock signal, NaCl 0.9%, 500 mM sodium phosphate buffer in D_2_O containing TSP (Trimethylsilylpropanoic acid) 0.05% for chemical shift calibration, and concentration reference, pH 7.0). Then, samples were centrifuged at 12,000× *g* for 10 min, and 600 μL aliquots of the supernatant were transferred into 5-mm NMR tubes for analysis. Analysis was performed using the ASCEND 800 MHz, AVANCE III HD Bruker spectrometer equipped with a 5 mm CPTIC 1H-13C/15N/DZ-GRD Z1194227/0011 cryogenic probe. The Carr–Purcell–Meiboom–Gill (CPMG) spin-echo pulse sequence (RD−90°-(τ-180°-τ) n-ACQ) was used as the nuclear magnetic resonance (NMR) sequence (CPMG condition: total T2 relaxation time of 60, 4 K data point, 128 scans, four dummy scans, delay time of eight seconds). 1D data obtained through the NMR analysis were subjected to baseline correction using the Chenomx program. Then, Binning was performed in units of 0.05 ppm, and spectral alignment was performed using the COW algorithm in MATLAB. Data organized by MATLAB were subjected to multivariate analysis with SIMCA −P++. 

For the quality control, TSP was used as an internal standard. We referenced the TSP peak to correct for chemical shifts and quantify metabolites. 

#### 4.1.3. Metabolite Pattern Analysis

To examine the difference in the metabolic pattern between the PD and HC groups, the 1H NMR spectrum was normalized to the total area and then orthogonal partial least-squares discriminant analysis (OPLS-DA) proceeded by UV scale. 

### 4.2. Gut Microbe Analysis

#### 4.2.1. Meal Adjustment Guide

The subjects were instructed not to drink alcohol or to eat excessively fatty foods the day before stool collection.

#### 4.2.2. Stool Collection and Specimen Delivery

Stool (4 mg) was sealed in a kit for stool collection. The outside of the kit was labeled, so that the subjects’ specimens could be distinguished. The specimens were then frozen at a temperature of −20 °C and delivered to the analytical laboratory.

#### 4.2.3. Gut Microbe Analysis

A library was created to enable Illumina sequencing by creating a hybrid primer that selectively amplified the V3–V4 region from the 16S rRNA gene, which is the standard for identifying bacteria, and an adaptor sequence recognized by the Illumina sequencer. The complete sequencing library mixture was sequenced by 300 bp paired-end sequencing according to Illumina’s MisSeq platform guide. After trimming the sequenced data, the quantitative insights into the microbial ecology (QIIME) pipeline was used to identify the bacteria. Greengenes was used as a library for bacterial identification. Analysis was conducted with a total of 20 samples that passed quality control. Alpha diversity, which compares the diversity distribution of gut microbes, was compared, and principal coordinate analysis (PCoA) was performed using the Bray–Curtis distance for pattern analysis. 

## 5. Statistical Analysis

The data collected from the subjects were coded and analyzed using SPSS for Windows (ver. 20.0) statistical program. To check the normality of the data, the Shapiro–Wilk test was used for continuous variables. An independent t-test or Mann–Whitney U test was used to compare the levels of blood metabolites and gut microbes in the stool between the PD and HC groups. To control confounding factors, independent *t*-tests or Mann–Whitney U test were used for both groups of the sex and age. *p* values < 0.05 was considered as statistically significant. 

## 6. Results

### 6.1. Subject Characteristics

From December 2019 to March 2020, 10 PD patients and 10 healthy individuals were recruited, and the characteristics of the subjects are presented in Table 1. The demographic characteristics including the sex and age did not show a significant difference between the two groups. 

### 6.2. Metabolite Analysis 

PCA (R^2^X = 0.629, Q^2^ = 0.187, Figure 1) and OPLS-DA (R^2^Y = 0.605, Q^2^ = 0.37, Figure 2) showed a clear differentiation of metabolites between the PD and HC groups. The established model was considered reliable according to the cross-validation with a 100-permutation test (Figure 3). Green R^2^- and blue Q^2^-values to the left were lower than the original points to the right, and the regression line of the Q^2^-points intersected the vertical axis below zero (R^2^ = 0.278, and Q^2^ = −0.220). The corresponding regression coefficients for the included metabolites, ordered according to their variable importance in the OPLS-DA model, are shown Figure 4. Among the metabolites analyzed, the levels of acetate, citrate, methionine, and trimethylamine were significantly different between the two groups (*p* < 0.05). Acetate and citrate levels were significantly higher, and methionine and trimethylamine levels were significantly lower in the PD group than in the HC group (Figure 5). 

### 6.3. Gut Microbe Analysis

Comparison of the alpha diversity between the two groups revealed that the PD group had significantly lower Chao1 levels, which indicates the diversity of gut microbes in this group compared to the HC group (*p* = 0.036).

Comparison of the results of the PCoA based on the Bray–Curtis distance showed that the two groups had different gut microbial patterns, but the patterns were not clear (Figure 6). 

In the distribution of gut microbe composition, *Bacteroidetes* in the phylum level showed a significant difference between the two groups. *Prevotella*, *Phascolarctobacterium*, *Pseudoflavonifractor*, *Eisenbergiella*, and *Gemella* at the genus level were also significantly different between the two groups (Table 2). Gut microbe composition at the phylum and genus levels of the PD and HC groups are shown in Figure 7 and Figure 8, respectively.

## 7. Discussion

We performed metabolite and gut microbial analyses in 10 PD patients and 10 healthy individuals and compared their results. Among the metabolites, the acetate, citrate, methionine, and trimethylamine levels were significantly different between the two groups. In the gut microbes, the abundance of *Bacteroidetes*, *Prevotella*, *Phascolarctobacterium*, *Pseudoflavonifractor*, *Eisenbergiella*, and *Gemella* were also significantly different. 

In this study, both acetate and citrate levels were higher in the PD group than in the HC group. Beynen et al. reported that acetate drastically increased the cellular level of citrate, which explains the results of this study [15]. 

Acetate enhances de novo lipid synthesis by activating lipogenic genes. That is, acetate is an immediate metabolic precursor and plays a role in inducing fatty acid synthesis [16]. Potashkin et al. reported that plasma oxidized LDL, a major contributor in atherosclerotic plaque formation in patients with PD, was high, which is in line with the results of this study, showing that PD patients had high acetate levels [17]. 

Citrate is an intermediate of the Kreb’s cycle. Pyruvate is an end metabolite of glycolysis, which is converted into acetyl-coA by an enzyme called pyruvate dehydrogenase (PDH) and enters the Kreb’s cycle. Citrate plays a role in regulating the activity of PDH [18]. In a 2019 study, it was reported that PD patients showed poor regulation of glucose metabolism in the substantia nigra pars compacta dopamine neurons [17]. Considering that citrate affects the Kreb’s cycle, a representative pathway of glucose metabolism, the results of this study are consistent with the 2019 study [17].

Methionine and trimethylamine levels were significantly lower in the PD group than in the HC group, which is also consistent with the results of previous studies. Glaser et al. reported that the accumulation of α-synuclein in dopaminergic neurons of the substantia nigra is a crucial step in the pathogenesis of PD, and methionine oxidation inhibits the accumulation of α-synuclein [19]. Andreasson et al. reported that the development of polyneuropathy in patients with PD was associated with disorders of methionine cycle metabolism [20]. 

According to previous studies, it was reported that trimethylamine N-oxide (TMAO) was associated with the severity and progression of motor symptoms in PD, and PD patients have a significantly lower TMAO level than healthy individuals [21]. In this study, TMAO was not significantly different between the two groups, but the trimethylamine levels were found to be significantly lower in the PD group. Considering that TMAO is produced by the oxidation of trimethylamine, the results of this study cannot be said to be contrary to the existing results [21]. 

Among the gut microbes, *Bacteroidetes*, which was significantly different between the two groups, are known to produce favorable metabolites such as short-chain fatty acids (SCFAs) that reduce inflammation (e.g., allergic inflammation) [22]. In addition, Aho et al. reported that patients with PD had different gut microbes, SCFAs, and inflammation from those of healthy individuals, which is consistent with the results of this study [23]. 

*Prevotella* produces hydrogen sulfide (H_2_S), which is a gasotransmitter that is closely related to neuroprotection as well as PD [8]. According to Kessel et al.’s gut microbe studies, the abundance of *Prevotella* showed a significant decrease in four out of 13 PD patients, and this study also showed a significant difference in *Prevotella* between the PD group and the HS group [8]. 

*Phascolarctobacterium* produces SCFAs and is known to be associated with the development of non-alcoholic fatty liver disease (NAFLD) [24]. Vural et al. reported that the prevalence of NAFLD in the PD group was significantly different from that of the HC group, consistent with *Phascolarctobacterium*, which showed a significant difference between the two groups in our study [25]. 

*Pseudoflavonifractor* is a gut microbe associated with energy metabolism and insulin sensitivity, which can exacerbate metabolic disorders in diabetic patients [26]. Hassan et al. reported that DM and PD are associated and have common mechanisms such as hyperglycemia, inflammation, and oxidative stress. Therefore, anti-diabetic drugs may have a beneficial effect on PD [27]. In our study, *Pseudoflavonifractor* showed a significant difference between the two groups, which is consistent with previous studies [26,27]. 

*Eisenbergiella* has been reported to be associated with major depressive disorder [28]. The significant difference in the abundance of *Eisenbergiella* between the two groups in our study may explain the reason why many PD patients experience depression [29]. This also means that drugs targeting *Eisenbergiella* can be used instead of antidepressants with various side effects. 

*Gemella* has been reported as a gut microbe associated with Alzheimer’s disease [30]. Considering that cognitive impairment is common in PD, a significant difference in *Gemella* between the two groups in our study is an acceptable result [31]. 

When synthesizing the previous studies above-mentioned, metabolites and gut microbes that were significantly different between the PD and HS groups in our study may be mainly related to lipid and glucose metabolism as well as inflammation (Table 3). Potashkin et al. also reported that PD patients exhibit abnormalities in lipid and glucose metabolism [17].

Sebastiaan et al. said that metabolites are the products of gut microbes [8]. Gibiino et al. reported that acetate production soon changes according to the change in *Firmicutes*/*Bacteroidetes* proportion [32]. This explains why metabolism involving metabolites and gut microbes, which showed significant differences between the two groups, were similar in our study. Additionally, this suggests that drugs that control gut microbes can also control metabolites and can have a positive effect in relieving symptoms in PD patients through a different route than the existing dopamine drugs. In another study supporting this hypothesis, Meng-Fei et al. reported that microbe-targeted interventions such as antibiotics, herbal medicine, probiotics, and fecal microbe transplantation could favorably affect PD [13]. 

Therefore, considering the results of this study and existing studies, an integrated approach is needed in the development of therapeutic drugs for PD because not only the brain but also the gut and diseases such as DM, hyperlipidemia, and PD must be considered. In addition, herbal medicine, probiotics, fecal microbe transplantation, and herbal medicines should be considered when selecting candidates for therapeutic drugs. 

The limitations of this study are as follows. First, as the present work is a pilot study, the number of subjects analyzed is small; thus, they are insufficient to reflect the characteristics of all patients with PD. However, the reliability of the results is not expected to be low, since significant results were derived from our study despite the small number of patients, and are consistent with the existing research results. Second, this study did not compare the differences based on the subjects’ detailed diet, the type of PD-related drugs they were taking, or their sex. However, it was the same in the broad framework of traditional Korean dishes and PD-related drugs such as levodopa and dopamine agonists. In addition, the male-to-female ratio between the two groups did not differ much. As a result, the importance of the findings of this study cannot be dismissed. Third, this study did not evaluate the potential correlations between the metabolites and gut microbes that showed a significant difference between the two groups. However, it was confirmed that they were commonly related to lipid and glucose metabolism and inflammation. Fourth, we could not suggest the names of gut microbes at the species level that showed a significant difference between the two groups. However, we were able to confirm the lack of gut microbe diversity at the species level in the PD group through an alpha diversity analysis, which evaluates the diversity of gut microbes at the species level. 

In conclusion, to the best of our knowledge, most of the existing studies have only analyzed either the metabolites or gut microbes. However, in this study, both metabolites and gut microbes were collected from the same subjects, measured, and compared, and our data confirmed that the metabolites and gut microbes that were significantly different between the PD and HS groups were mostly related to lipid and glucose metabolism. If significant differences are confirmed through large-scale studies comparing gut microbes and metabolites before and after various treatments such as herbal or Western medicine, diet, and feces transplantation, the results will be helpful in the development of new therapeutic drugs for PD. 

## Figures and Tables

**Figure 1 healthcare-10-00302-f001:**
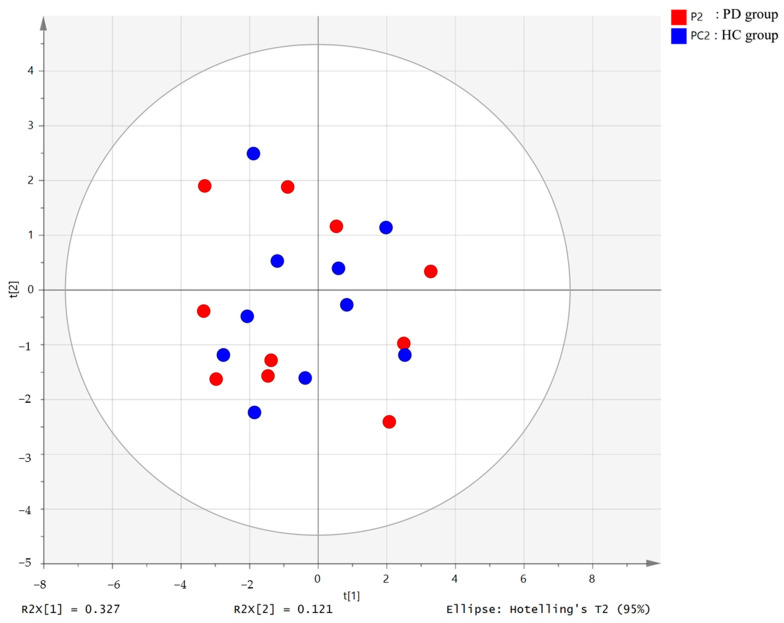
PCA score plot derived from the 1H-NMR spectra of serum from the Parkinson’s disease (PD) patient group (*n* = 10) and healthy control (HC) group (*n* = 10). PD, Parkinson’s disease; HC: healthy control.

**Figure 2 healthcare-10-00302-f002:**
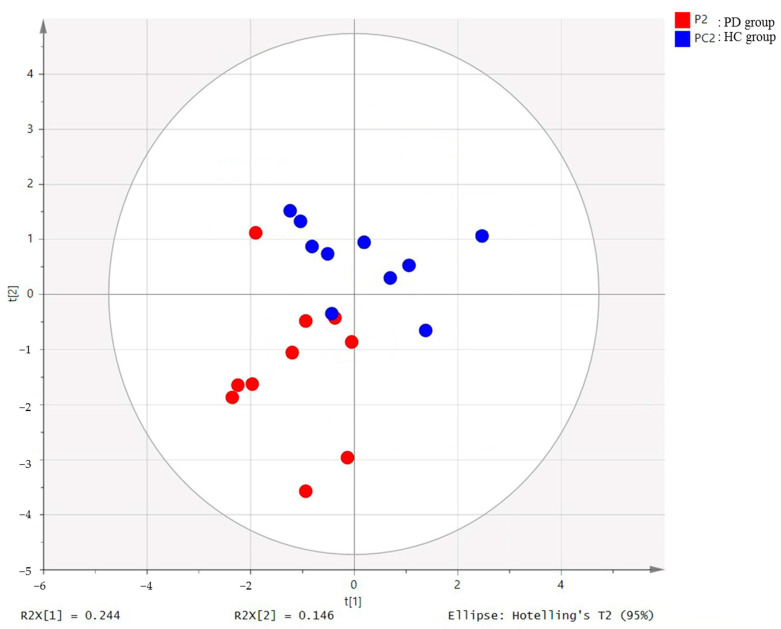
OPLS-DA score plot derived from the 1H-NMR spectra of serum from PD (*n* = 10) and HC groups (*n* = 10). OPLS-DA, orthogonal partial least-squares discriminant analysis; PD, Parkinson’s disease; HC: healthy control: NMR, nuclear magnetic resonance.

**Figure 3 healthcare-10-00302-f003:**
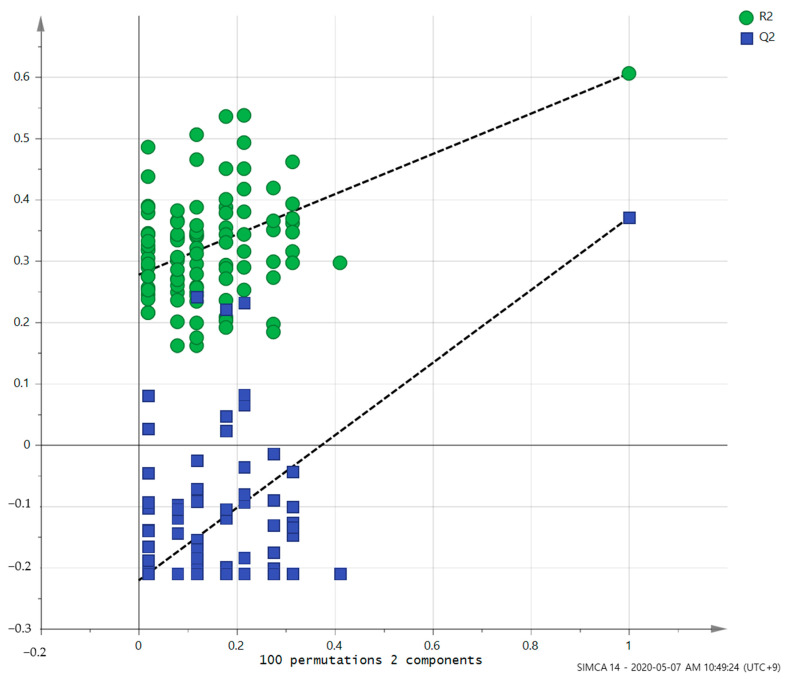
Validation of the OPLS model using the 100-permutation test.

**Figure 4 healthcare-10-00302-f004:**
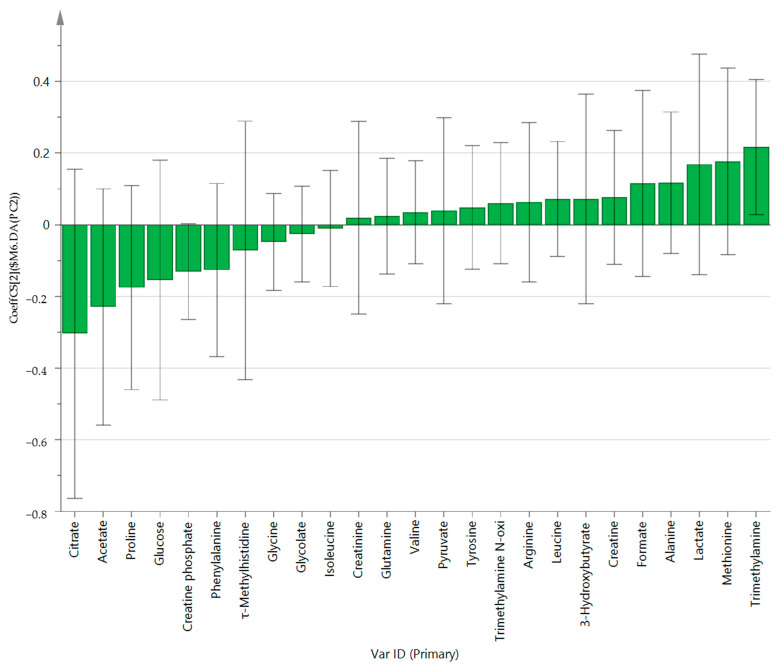
OPLS-DA coefficient plot of all metabolites in Parkinson’s disease patients.

**Figure 5 healthcare-10-00302-f005:**
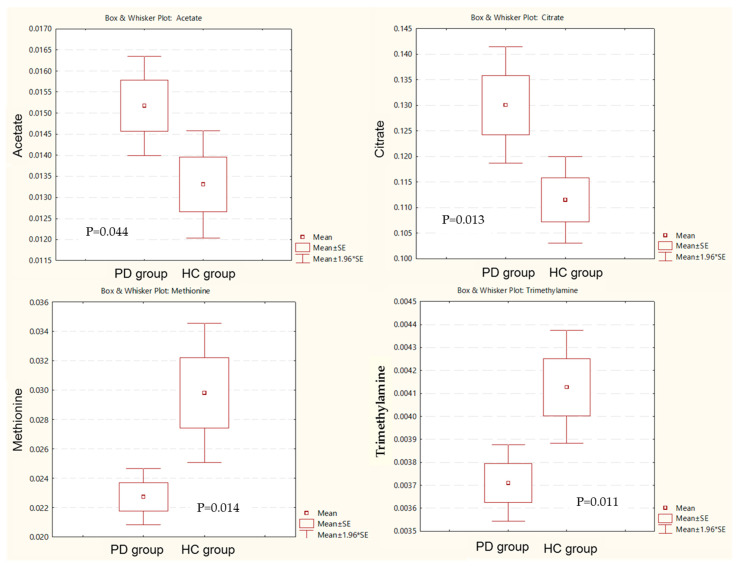
Box and whisker plot of acetate, citrate, methionine, and trimethylamine in the Parkinson’s disease (PD) group and healthy control (HC) group. PD, Parkinson’s disease; HC: healthy control.

**Figure 6 healthcare-10-00302-f006:**
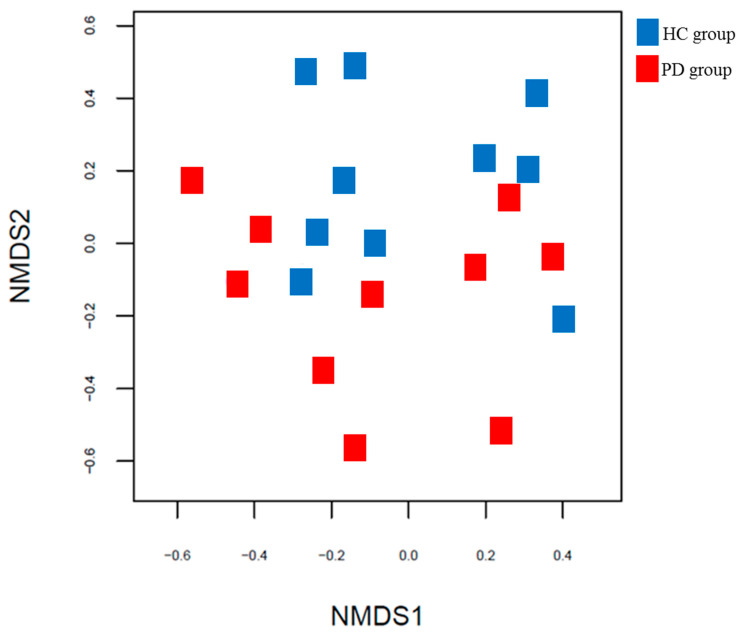
PCoA plots based on Bray–Curtis distances between the PPD and HC groups. PCoA, principal coordinate analysis (PCoA); Parkinson’s disease; HC: healthy control.

**Figure 7 healthcare-10-00302-f007:**
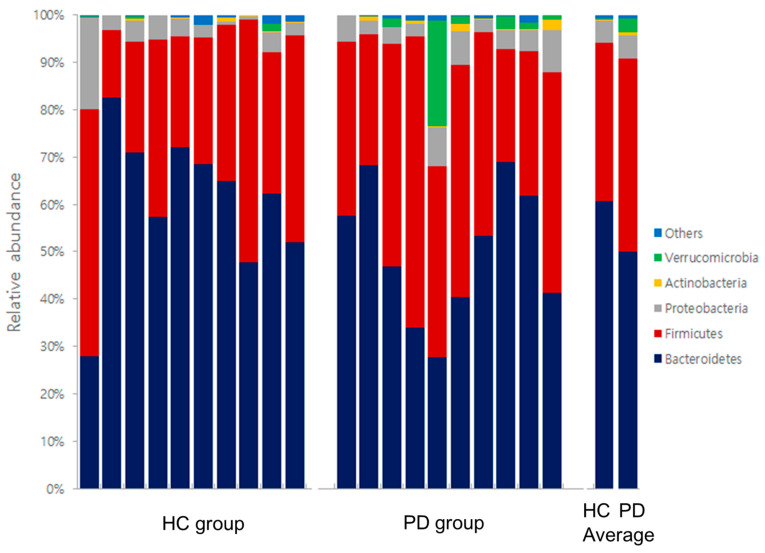
Gut microbe composition at the phylum levels of the HC and PD groups. HC: healthy control; PD: Parkinson’s disease.

**Figure 8 healthcare-10-00302-f008:**
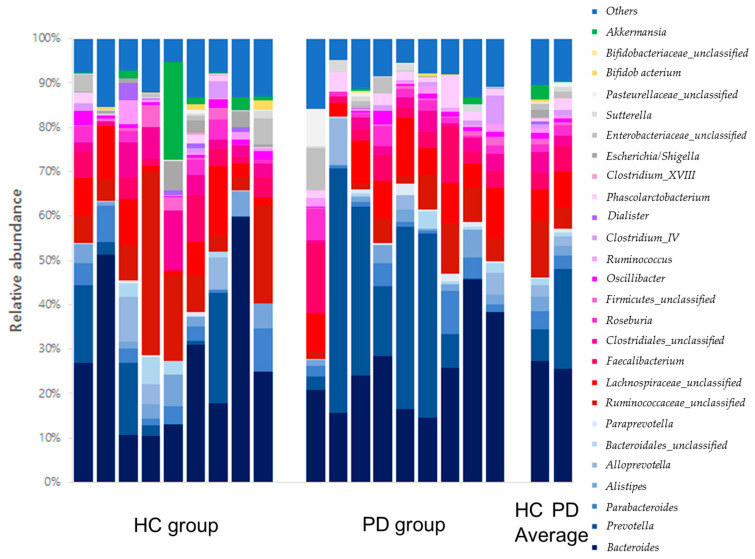
Gut microbe composition at the genus levels of the HC and PD groups. HC: healthy control; PD: Parkinson’s disease.

**Table 1 healthcare-10-00302-t001:** Demographic characteristics and medical history of the enrolled subjects.

Classification		PD Group	HC Group	Total
Total		10	10	20
Sex (number)	Male	5	4	9
	Female	5	6	11
Age (years)	Minimum	50	51	50
	Maximum	74	68	74
	Average	66.6	58.7	62.65
Disease duration (years)	Minimum	0.17	-	0.17
	Maximum	12	-	12
	Average	5.02	-	5.02

PD, Parkinson’s disease; HC, healthy control.

**Table 2 healthcare-10-00302-t002:** Gut microbes that were significantly different between the PD and HC groups in this study.

Classification		Gut Microbes	PD Group vs. HC Group	
			↑/↓ ^§^	Significance (*p* < 0.05)
Stool	Phylum level	*Bacteroidetes*	↓	0.012
	Genus level	*Prevotella*	↓	0.030
		*Phascolarctobacterium*	↓	0.047
		*Pseudoflavonifractor*	↓	0.037
		*Eisenbergiella*	↑	0.037
		*Gemella*	↓	0.014

PD, Parkinson’s disease; HC, healthy control. ^§^ Arrows (↑ and ↓) indicate a decrease or an increase in the microorganism levels in patients with PD compared with healthy individuals.

**Table 3 healthcare-10-00302-t003:** Summary of metabolites and gut microbes that were significantly different between the PD and HC groups in this study.

Classification		Function
Metabolites	Acetate	Acetate is involved in lipid metabolism. It enhances de novo lipid synthesis by activating lipogenic genes [16].
	Citrate	Citrate is involved in glucose metabolism. As an intermediate of the Kreb’s cycle, it regulates the activity of PDH, an enzyme that converts pyruvate to acetyl-coA [18].
	Methionine	Methionine is involved in the accumulation of α-synuclein. The accumulation of α-synuclein in the dopaminergic neurons of substantia nigra is a crucial step in the pathogenesis of PD [19].
	Trimethylamine	Trimethylamine is oxidized to produce TMAO. TMAO is related to the severity and the progression of motor symptoms of PD [21].
Gut microbes	*Bacteroidetes*	*Bacteroidetes* produces favorable metabolites such as SCFAs that reduce inflammation [22].
	*Prevotella*	*Prevotella* produces H_2_S, which is a gasotransmitter closely related to neuroprotection [8].
	*Phascolarctobacterium*	*Phascolarctobacterium* has beneficial effects by producing SCFAs. It is known to be associated with the occurrence of NAFLD [24].
	*Pseudoflavonifractor*	*Pseudoflavonifractor* is associated with energy metabolism and insulin sensitivity [26].
	*Eisenbergiella*	*Eisenbergiella* is reported to be associated with major depressive disorder [28].
	*Gemella*	*Gemella* is reported to be associated with Alzheimer’s disease [30].

H_2_S: hydrogen sulfide; NAFLD, nonalcoholic fatty liver disease; PDH, pyruvate dehydrogenase; SCFAs, short-chain fatty acids; TMAO, trimethylamine N-oxide; PD, Parkinson’s disease; HC, health control.

## Data Availability

The data in this study are available from the corresponding author, S.L., upon reasonable request.

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
