# Peer review of "Comparison of Metabolites and Gut Microbes between Patients with Parkinson’s Disease and Healthy Individuals—A Pilot Clinical Observational Study (STROBE Compliant)"

_healthcare, 2022, doi:10.3390/healthcare10020302_

Round 1
Reviewer 1 Report
Kim et al., in their manuscript entitled "Comparison of Metabolites and Gut Microbes Between Patients with Parkinson’s Disease and Healthy Individuals —A Pilot Clinical Observational Study (STROBE compliant)" describe the significant alterations in metabolites and gut bacterial community in the PD cohort.
Major Concerns:
- Methods: The author did not mention whether they performed a metabolite extraction procedure. Please clarify and describe in details.
- Bacterial community analysis data: Please include more details. A figure depicting the Operational Taxonomic Unit (OTU) values of the overall 16S rRNA sequencing data is desirable for the improved readership.
- Same goes for the metabolite panel analysis: The authors need to add more information in addition to Figures 1 and 2. Using a comprehensive figure please highlight the top hits among the rest of the metabolites identified in the assay.
- Discussion: I think it will be helpful to the interested readers if the authors include a few names of the species, if any, (belonging to the bacterial families that are significantly different in this study) from previously published reports.
Minor concerns:
- Figure 2: Some of the fonts are too small to read.
- Figure 3: Please use separate colors too for the groups.
Author Response
We thank you very much for your detailed review and have actively revised the manuscript accordingly. The journal has not yet approved us to upload the revised manuscript, so we have added the revised figure to this file.
Please see the attachment.

Reviewer 2 Report
The study performed by Cheol-Hyun Kim and colleagues aimed at comparing the metabolites and gut microbes between patients with Parkinson’s disease (PD) and healthy individuals. The authors find different metabolic alterations between Parkinson’s disease and healthy individuals, as well as changes in alpha diversity and microbiota composition between groups. The topic of the paper is attractive and timely considering the growing evidences proposing for a role of the gut microbiota in different psychiatric conditions, including Parkinson’s disease. However, there are important issues that need to be addressed:
1- The inclusion and exclusion criteria for both groups are not completely clear. In the method section it is stated that some patients that “… were judged to be unsuitable to participate in this study by medical staff were excluded from the study”. This criterion is vague and needs clarification.
2- The demographic characteristic and medical history only account for sex, age and disease duration. No information is provided regarding the type of PD-related drug used among individual, type of diet, history of alcohol consumption, other psychiatric condition, antibiotic usage as well as other factors that are known to alter the microbiome. Similarly, no information regarding potential differences in metabolites or microbiota composition due to sex is mentioned. A more comprehensive analysis would strengthen these results.
3- The metabolomic analysis is very incomplete and lack validations. No data regarding the PCA analysis is shown. Similarly, there is no validation for the OPLS-DA model.
4- No information about the quality controls in the NMR analysis is provided.
5- Figure 1: in the OPLS-DA score plot there are two samples that seems to be significantly different from the rest (PC2-10 and PC2-05). Was any outlier detection method applied to the data? The presence of outlier can really impact the results of the PCA.
6- In the methods section, the authors mention that both, plasma and serum were collected. However, only data form plasma is shown. Is it any rationales behind the use of plasma over serum?
7- Figure 2: Individual data point would help on the observation and interpretation of the data.
8- The gut microbiome analysis performed in Figure 3 is rather superficial and provide very little information.
9- Table 2: presenting these data as a % of abundance would really facilitate the interpretation of the results and will allow for a more general observation of the overall changes in the microbiota composition between the groups.
10- Did the authors evaluated for potential correlations between the metabolic and microbial changes? While correlation does not mean causation, it would be interesting to see whether there is some potential interactions.
Author Response

(The authors gave the same response as above.)

Round 2
Reviewer 1 Report
This reviewer is thankful to the authors for addressing the concerns.
A few Points:
- Figure 7: I think, in line with the order shown in figure, it will better if the caption is written as "Gut microbiome composition at Phylum levels of the HC and PD groups ...."
- Figure 8: The X axis labels are missing. I assume they will be same as Figure 7. Please follow the same (as comment# 1) suggestion for the figure caption. The data and texts in the figure are blurry. Please include higher resolution figure.
- I am not sure this concern was addressed well - "Point 4: Discussion: I think it will be helpful to the interested readers if the authors include a few names of the species, if any, (belonging to the bacterial families that are significantly different in this study) from previously published reports." - the microbial information presented in table 3 are represented as alterations in bacterial families. I was wondering if the authors, in the discussion section, could talk about these changes in bacterial families (their own study) in the context of changes at the species level (from the existing literature).
Author Response
We’re really sorry to bother you. Thanks again for your detailed review, and the manuscript has been revised in full acceptance of your comments. Please see the attachment.
